# Ictal MEG-EEG Study to Localize the Onset of Generalized Seizures: To See Beyond What Meets the Eye

**DOI:** 10.3390/brainsci15090938

**Published:** 2025-08-28

**Authors:** Valentina Gumenyuk, Oleg Korzyukov, Noam Peled, Patrick Landazuri, Olga Taraschenko, Sheridan M. Parker, Darya Frank, Spriha Pavuluri

**Affiliations:** 1Department of Neurological Sciences, University of Nebraska Medical Center, Omaha, NE 68131, USA; okorzyukov@unmc.edu (O.K.); olha.taraschenko@unmc.edu (O.T.); sheparker@nebraskamed.com (S.M.P.); sppavuluri@childrensnebraska.org (S.P.); 2MGH/HST Martinos Center for Biomedical Imaging, Charlestown, MA 02129, USA; npeled@mgh.harvard.edu; 3Department of Neurology, University of Kansas Medical Center, Kansas City, KS 66160, USA; 4Andrew Mayes Centre for Cognitive Neuroscience, The University of Manchester, Manchester M13 9PL, UK; darya.frank@manchester.ac.uk; 5Department of Pediatrics, University of Nebraska Medical Center, Omaha, NE 68131, USA

**Keywords:** generalized seizures, MEG-EEG, brain source reconstruction

## Abstract

**Introduction:** Patients with generalized epilepsy are rarely referred for advanced diagnostics like magnetoencephalography (MEG). This is due to the assumption that generalized seizures cannot be localized noninvasively. **Methods:** We present simultaneous MEG (306 channels) and EEG (64 channels) data from seven patients with drug-resistant generalized epilepsy. Three patients experienced typical generalized seizures during their MEG clinical evaluation. In total, 38 epileptiform events (three seizures, 35 interictal discharges) were analyzed using two software platforms and three localization methods: equivalent current dipole (ECD), sLORETA (via SWARM), and dynamic statistical parametric mapping (dSPM). Individual head models were created from each patient’s MRI. **Results:** MEG successfully localized seizure onset zones, showing distinct hypersynchronous discharges on all sensors as well as alternately during interictal discharges. Localization was consistent across methods and generalized events within subjects, revealing cortical sources in all cases, with rapid propagation (27–60 ms) across networks. **Conclusions:** This study demonstrates that MEG can meaningfully localize both seizures and interictal discharges in generalized epilepsy. This supports a broader use for MEG beyond focal epilepsy. Incorporating MEG in drug-resistant cases including generalized epilepsies may improve diagnosis and guide treatments including non-surgical options.

## 1. Introduction

Generalized epilepsy is a common epilepsy syndrome, accounting for approximately 20% of diagnosed epilepsy in children and adults [1]. Patients with generalized tonic–clonic seizures are at higher risk for serious injury and sudden unexpected death in epilepsy (SUDEP) [2,3,4]. These seizures are electrophysiologically characterized by hypersynchronous discharges propagating across both hemispheres within milliseconds. EEG patterns typically consist of bilateral synchronous 2.5–5.5 Hz spike-and-wave or polyspike generalized discharges. Some seizure types also involve loss of consciousness which increases patient risk.

The International League Against Epilepsy (ILAE) defines generalized epilepsies as disorders with generalized seizures and interictal discharges but acknowledges that onset may be focal with rapidly engaged bilateral networks. Electrophysiological studies suggest that generalized activity can arise from complex networks involving both cortical and subcortical structures. However, the neurobiological origin of generalized seizures—especially absence seizures—remains unknown [5]. In 2001, the ILAE has stated that “previously used classifications based on the concepts of ‘partial’ vs. ‘generalized’ created the false impression that epileptic seizures or syndromes could be either due to localized disturbances in one hemisphere or disturbances involving the entire brain. In fact, there are a variety of conditions between focal and generalized epileptogenic dysfunctions, including diffuse hemispheric abnormalities” [6].

To avoid misinterpretation between idiopathic generalized epilepsy (IGE) and focal seizures, the ILAE recommends not strictly categorizing seizures and epilepsy syndromes as one or the other, since in some patients both focal and generalized forms of epilepsy may coexist. Based on this recommendation from the ILAE, high-density EEG (HEEG), MEG, or both methods together may serve as valuable diagnostic tools to clarify the “focality” feature in all epilepsies, including generalized epilepsies, whether primary or secondary (focal).

Multiple theories for generalized epilepsy have been proposed such as thalamic pacemaker models and the cortical focus theory which posits that seizures may begin in focal cortical regions, especially the frontal lobes, and then rapidly spread through cortico-cortical pathways [7,8,9,10,11,12]. However, rapid propagation in focal epilepsy raises concerns about misdiagnosis. Advanced imaging modalities with high temporal and spatial resolution are needed to distinguish true generalized onset from rapid focal spread [13]. The cortical spread theory does not exclude the participation of the thalamus in generalization but emphasizes the thalamus’ role as a secondary generator.

Several scalp and intracranial EEG studies have shown focal spike-wave onsets in generalized epilepsy patients, providing further support for the cortical focus theory [14]. According to this model, spike-wave discharges are consistently delayed in other brain regions relative to a specific focal onset site with propagation occurring within a few milliseconds [7,14,15,16].

Standard scalp EEG has limitations in localizing or lateralizing generalized epileptiform activity. In contrast, MEG offers both millisecond temporal resolution and millimeter spatial resolution. This makes MEG well-suited to localize epileptiform sources and their rapid propagation. MEG is a noninvasive tool used in presurgical evaluations, particularly in focal epilepsy; still, it is rarely applied to generalized epilepsy due to the assumption that such seizures cannot be localized. Hughes et al. [17] were among the first to report MEG utility in generalized epilepsy, emphasizing its value beyond scalp EEG alone [18]. Utilizing advanced signal processing techniques, such as high-density EEG and MEG, researchers can record and localize interhemispheric latency differences of up to 10–20 ms with accuracy within 5 mm [10,19,20].

MEG allows for advanced source reconstruction techniques such as equivalent current dipole modeling, beamforming, and distributed source models like dSPM and standardized LORETA (sLORETA) [21]. These methods help solve the inverse problem which estimates the location and orientation of brain activity sources based on recorded signals. While MEG is highly sensitive to tangential sources near the cortex, it may miss deeper or radial sources. Therefore, combined MEG-EEG recordings are particularly valuable to ensure recording of both tangential and radial sources [22].

Recent studies support the clinical use of simultaneous MEG and EEG for enhanced localization, yet few have applied this approach to generalized epilepsy. Integrating electric and magnetic signals (electromagnetic source imaging) can improve seizure onset localization and evaluation of rapid propagation more precisely [18,23]. Generally, maximal clinical information is achieved when simultaneous MEG and EEG data are subjected to source modeling, either individually or in a combined manner [24].

In this study, we present simultaneous MEG and EEG data from seven patients with drug-resistant generalized epilepsy. We aim to determine whether seizures and interictal discharges in these cases originate from consistent focal regions and that they have rapid propagation as suggested by the cortical focus theory.

## 2. Materials and Methods

### 2.1. Patients

Seven patients (ages 12–33, five males) with generalized seizures and potential generalized epilepsy (Table 1) were referred for clinical MEG examinations at the University of Nebraska Medical Center (UNMC), a level 4 epilepsy center. Prior to MEG study, all patients underwent 3T MRI study, long-term video monitoring with scalp EEG, and neuropsychological evaluation. Six patients also completed a PET study prior to the MEG exam. Scalp EEG studies confirmed that generalized discharges and seizures were present without clear localization or lateralization (Table 1). Clinical MEG examinations and a review of each patient’s clinical history were conducted in compliance with the UNMC’s clinical and research policies. The research protocols were approved by the UNMC Committee on Human Research (IRB #0714-21-EP).

### 2.2. MEG/EEG Data Acquisition Analysis

Simultaneous MEG/EEG data were recorded using a 64-channel EEG cap (EasyCap, Brain Vision, Gilching, Germany) and a 306-sensor MEG system (TRIUX neo, MEGin, Helsinki, Finland) in a magnetically shielded room. Data were acquired at a 1000 Hz sampling rate with a 0.03–300 Hz bandpass filter. Patients reduced their total sleep time by ~40% the night before by delaying sleep onset for 2–3 h. During recording, patients laid supine with head support in the MEG helmet and reached stage 2–3 sleep.

Head position was tracked using five head position indicator (HPI) coils placed on the EEG cap. Coil locations were digitized relative to anatomical landmarks using a 3D digitizer (Polhemus, Colchester, VT, USA). All patients underwent a clinical T1-weighted brain MRI for anatomical reference.

Both MEG and EEG data were recorded for six 10 min sessions. Three patients (patients #1, #2, and #3) experienced typical generalized tonic–clonic seizures during MEG/EEG recording. The onset of each generalized seizure was captured during recording and analyzed offline. All patients were postictally safe and were able to continue with the MEG clinical study.

### 2.3. MEG/EEG Data Analysis

MEG/EEG data were processed offline with artifact removal performed by spatially filtering the raw data using the temporal extension of Signal Space Separation (tSSS), as implemented in the MaxFilter software [25]. Brain waveforms were independently reviewed for the presence of epileptiform activity including generalized seizure onset and interictal generalized discharges by an epileptologist (S.P.) and neurophysiologist (V.G.).

Epochs corresponded to the seizure along with 2 s prior to onset and 2 s following the evolution of the seizure were selected as pre-onset and post-onset epochs. In addition, 5–10 s epochs of interictal generalized discharges were selected for each patient which included 2 s of activity before and after the discharge.

For patients #1–3 that experienced generalized seizures during MEG/EEG, 4 s seizure epochs and 5–10 interictal epochs were selected. For the remaining four patients, 5–10 epochs of 4 s interictal generalized discharges were analyzed.

Two software tools were used for source localization: CURRY (Compumedics, Australia) and MMVT [26]. CURRY is FDA-approved and supports both ECD and sLORETA methods [21,24,27] while MMVT enables integration of MEG and EEG data, normalization to sleep baselines, and visualization of propagation patterns. Using both tools ensured robust clinical interpretation and data validation.

For distributed source modeling, we applied sLORETA which is known for its accuracy and validated in CURRY [21,24,27]. Sleep normalization was performed in MMVT using each patient’s MEG sleep baseline to distinguish epileptogenic activity from normal stage 2 and stage 3 sleep patterns (see [19]).

In CURRY 7.0, seizure and interictal discharge onsets were visually marked by an epileptologist (SP) and a neurophysiologist (VG) independently. Seizure and interictal discharge onsets were identified as the first significant deflection from baseline in the MEG/EEG traces. These markings corresponded to ictal and interictal individual epochs clipped from the raw continuous 10 min data. Using this visually marked time point (corresponding to 0 time in our study), the distributed source-reconstruction–brain-localization analysis was performed. Data were filtered with a 60 Hz notch and 0.5–70 Hz bandpass filter. Signal-to-noise ratios were calculated for each epoch. We used the automatic option implemented in CURRY 7.0 which detects and applies the most appropriate method for noise estimation. For our continuous data, this corresponded to the Percentile 20 method which calculates the standard deviation using the smallest 20% signal values across the entire latency range for noise estimation. Electrode positions were co-registered to the MRI-based 3D head model using predefined coordinates. Onset events were free of artifacts from cardiac or eye activity.

The same ictal and interictal epochs, visually marked for onset, were submitted to MMVT. We used MMVT to refine the identification of seizure and interictal discharge onsets by applying a z-score approach in addition to visual detection. This allowed our subjective definition of ictal and interictal onsets to be verified with an objective z-score. MMVT applies surface-based source estimation (e.g., dSPM) across 4000 cortical points per hemisphere. Results were normalized to each patient’s sleep activity, yielding z-values from 2 s before to 2 s after onset, reflecting epileptogenic activity relative to normal sleep [19]. Sleep-state recordings provided a consistent baseline for all patients.

### 2.4. Statistical Analysis

To identify seizure and interictal discharge onsets, we calculated the maximum z-value for each hemisphere and their difference. A focal onset was assumed if a z-value peak appeared in only one hemisphere. MMVT displayed a baseline band (cyan) defined as mean ± 2 SD of the −0.5 s to 0 s pre-onset interval. Under a Gaussian assumption, this band approximates a 95% interval. Samples exceeding it (|z| > 2) lie outside the baseline’s ~95% range. We treated |z| > 2 relative to the −0.5 s to 0s baseline as exceeding the baseline’s ~95% interval and then required a significant interhemispheric difference (cluster-based permutation [28]). The segment meeting both criteria nearest to the manual mark was retained; the onset was the peak within that segment (dashed red line). Z-value propagation was visualized in 10 millisecond windows, based on prior studies showing epileptic spike spread over this timescale [28]. Reproducibility was assessed qualitatively by the spatial concordance of multiple events within each subject (ictal vs. interictal overlap and clustering across repeated interictals).

## 3. Results

The clinical evaluation results of the study patients are shown in Table 1.

The imaging (MRI/PET) and long-term EEG results were not concordant and were unable to lateralize or localize the generalized seizures and generalized interictal discharges. Patients exhibited both focal and generalized features on scalp EEG with some patients describing focal semiology during their seizures. These findings, combined with their drug resistance, prompted further investigation. Additionally, it is notable that MRI showed no abnormal areas in five of the seven patients. Importantly, patients #1, #2, and #3 experienced a clinical seizure during the MEG/EEG recordings. The onset of the recorded generalized seizure traces, along with the MEG/EEG localization of seizure onset and propagation, are shown in Figure 1. From these figures, we can see that the propagation of seizure activity is patient specific. Patient #1 had more generalized onset involvement at 27 ms latency whereas patients #2 and #3 had reduced activity at generalized onset but instead had increased activity in adjacent areas.

### 3.1. Seizure Onset and Propagation Defined by Z-Values in Normalized Epileptogenic Data to Normal Sleep Data

The results of generalized seizure onset localization based on the z-value plot corresponding to sleep-ictal normalized data for all patients are shown in Figure 2. Using MMVT, we found the manually identified onsets corresponded to the timing of the peak activity (*p* < 0.02) in all three patients. Across events within each patient, onsets converged temporally near 0 s and spatially within the same frontal gyrus (see Figure 2B), indicating stable, within-subject localization. As shown in Figure 2A, the z-value was significantly higher with respect to the baseline (highlighted in cyan): −12 ms, −6 ms, and −8 ms for the three patients, respectively. The confidence interval was calculated based on the maximum z-values of both hemispheres before time 0 s, where the higher limit (mean + 2-std) was 6.3, 5.8, and 4.3 for the three patients, respectively.

Importantly, for this analysis we normalized epileptogenic activity against normal sleep activity (free of epileptic activity) to generate z-values and remove normal sleep activity. Therefore, these plots depict the z-values for the left and right hemispheres as an objective illustration of the time corresponding to seizure onset and the peak of normalized activity corresponding to seizure onset without any other neurophysiological brain signal. The peak has a sharp tip which suggests the true onset of generalized activity in the cortical areas (precentral in Pt#1, superior-frontal in Pt#2, and rostral-middle-frontal in Pt#3).

The spatiotemporal propagation of generalized seizures in the patients is shown in Figure 3 and Figure 4. The propagation was calculated from onset to 50 ms and was colored using a yellow-to-red gradient where the onset was in red and yellow represented a later time (Figure 4).

### 3.2. Interictal Generalized Discharges

Figure 3 shows interictal generalized activity traces for all seven patients illustrating that each epoch reflects patient-specific patterns. EEG onset appeared more distributed compared to MEG (left panel). In patients #1, #2, and #3, interictal onset locations closely matched ictal event onset regions shown in Figure 1.

Figure 4A displays onset localization of interictal discharges based on z-value plots from sleep-normalized dSPM estimates in these three patients. Each interictal event was analyzed individually, aligned to manually identify onset, and normalized to sleep baseline. Peak z-values were significant (*p* < 0.02) and occurred near 0 s (−12 ms, −11 ms, and −5 ms). Corresponding confidence interval upper bounds were 3.27, 2.68, and 2.53, with peak localizations at 27 ms, 59 ms, and 45 ms.

Figure 4B shows spatiotemporal propagation over 50 ms with the onset marked in red and later activity in yellow, similar to the ictal event mapping.

### 3.3. Comparison Between Ictal and Interictal Source Localization Results

Within each of the three seizure cases, interictal onset maps overlapped the ictal onset region (Table 2; Figure 3 and Figure 4), supporting within-subject reproducibility. It revealed frontal lobe focality in both ictal and interictal MEG/EEG data. Seizure propagation varied by patient with subcortical areas (patient #1), bilateral frontal lobes (patient #2), and premotor cortex (patient #3) spread occurring within 27–60 ms of onset.

Interictal discharge localization involved frontal regions in most patients (86%) with patient #6 showing activity in both frontal and parietal lobes.

Across 38 epochs (3 seizures, 35 interictals), combined EEG-MEG analysis consistently localized onset to cortical regions. No thalamic sources were identified supporting the cortical focus theory [14].

## 4. Discussion

We sought to examine the onset of both generalized seizure and generalized interictal discharge locations using EEG and MEG data from seven patients with generalized epilepsy. Our findings suggest people with generalized epilepsy can have localizable onsets for both interictal and ictal findings. The authors of this study were motivated to combine facets of the cortical focus theory and the ILAE’s suggestion of focal features in the onset of generalized epilepsy. However, to date, it is still common practice, perhaps even typical, that patients with generalized epilepsies are not referred for tests providing additional localization. Our findings suggest patients with generalized epilepsy may benefit from combined EEG and MEG localization analyses. The integration of available neuroimaging tools, particularly MEG, offers potential advantages for enhancing diagnostic precision and optimizing treatment planning. MEG, as a clinical tool, may be especially valuable in cases where patients present with focal semiology, but are otherwise considered to have generalized epilepsy. Furthermore, MEG’s ability to noninvasively localize could be increasingly important in the expanding field of neuromodulation.

Regarding variability and reproducibility within subjects, repeated events clustered to a single gyral region and ictal onsets overlapped interictal onsets (Figure 2, Figure 3 and Figure 4), supporting reproducibility without re-averaging. Between subjects, onsets localized to the right frontal cortex with subregional differences emerging only during propagation (>50 ms). These patterns align with prior MEG/EEG reports of gyral-scale variability while remaining within clinically actionable spatial tolerances for intracranial-EEG targeting.

Despite the small number of patients in our study, the results are aligned with the cortical focus theory assertion that onset of generalized epileptic activity lies within focused cortical areas, with subsequent rapid propagation (within tens of milliseconds) to subcortical regions as demonstrated in previous research (see review [7,13]). Accurate localization techniques, such as combined MEG and EEG, recordings can more precisely guide treatment approaches, including neuromodulation for patients with generalized epilepsies. Ongoing clinical trials investigating brain-responsive neurostimulation have shown promising results by targeting the thalamocortical network in patients with one of the generalized types of epilepsy such as a Lennox–Gastaut Syndrome or idiopathic generalized epilepsy. However, incorporating precise localization of seizure onset may further enhance effectiveness of neurostimulation by enabling co-implantation within specific parts of the epileptogenic network rather than relying solely on thalamocortical pathways. In our study of seven patients spanning 38 epileptiform epochs (three seizures and 35 interictal discharges), none were localized to the thalamic areas at their onset. For each epoch localization analysis, EEG data were used with MEG data to ensure that the results were not limited to MEG signals alone [29]. Studies have shown that MEG and EEG are complementary techniques for investigating the full spectrum of human brain activity as MEG can detect neural activity that EEG cannot detect and vice versa [30]. Compared to EEG, MEG provides higher spatial resolution for most focal neocortical sources and a lower signal-to-noise ratio for extended sources [31]. The results of the localized onset of generalized seizures in our study demonstrated that the combined MEG and EEG method is both useful and effective in detecting the brief massive deflection of brain activity that rapidly progresses to a generalized pattern.

A difference in the pathogenesis between ictal and interictal discharges has been demonstrated through computational analysis of functional connectivity [32]. This study revealed two distinct networks in interictal generalized discharges: one responsible for synchronized interictal activity at 3–4 Hz (delta) and the other for desynchronizing interictal activity at 8–10.5 Hz (alpha). Anatomically, the nodes influencing the interictal state are primarily localized within the default mode network, dorsal attention network, visual network, and thalamus. In our MMVT analysis, we demonstrated rapid propagation within both the networks underlying seizure activity and interictal discharges in the data normalized to sleep. Therefore, the use of tools to accurately delineate seizure onset from seizure propagation is of paramount importance. In this study, we demonstrate the utility of combining precise tools and advanced analyses for the evaluation of generalized epilepsies to address this need.

Focal epilepsies have greatly benefited from advanced neuroimaging and precise surgical treatment options; however, such options have remained elusive for idiopathic generalized epilepsy syndromes. Relying on broad-spectrum anti-seizure medications continues to be the mainstay treatment option in these patients with escalation to broad-network neuromodulation options like vagus nerve stimulation in drug-resistant cases. Understanding onset and propagation patterns can provide targeted treatment options for these patients. Early studies in drug-resistant childhood absence epilepsy (CAE) have already shown focal activity prior to generalized spread, leading to the understanding of focused seizure generation location for generalized epilepsies even in early ages [33]. These early investigations have further led to the discovery of seizure-detection devices, like the eye-tracker glasses, resulting in early and accurate diagnosis of CAE [34]. Efforts to accurately diagnose and treat these epilepsy syndromes with the aim to reduce morbidity are already underway (e.g., NCT06310772) and our study further catapults these findings to the next research stage with expansion to include drug resistant generalized epilepsy patients in addition to the decades of work in focal epilepsies.

## 5. Conclusions

Our study provides results-driven evidence supporting a cortical focus at the onset of both ictal and interictal generalized activities. Using two different programs with distinct methodologies and combining subjective and objective analyses, we assessed whether generalized epileptogenic activity could be localized to the thalamic region. Whole-head EEG data were analyzed alongside MEG data with no onset localized to the thalamic regions. While we acknowledge the small sample size, we hope these findings encourage further localization research for generalized epilepsies using noninvasive methods to improve diagnosis and guide treatment planning for patients, particularly those that are drug resistant.

## Figures and Tables

**Figure 1 brainsci-15-00938-f001:**
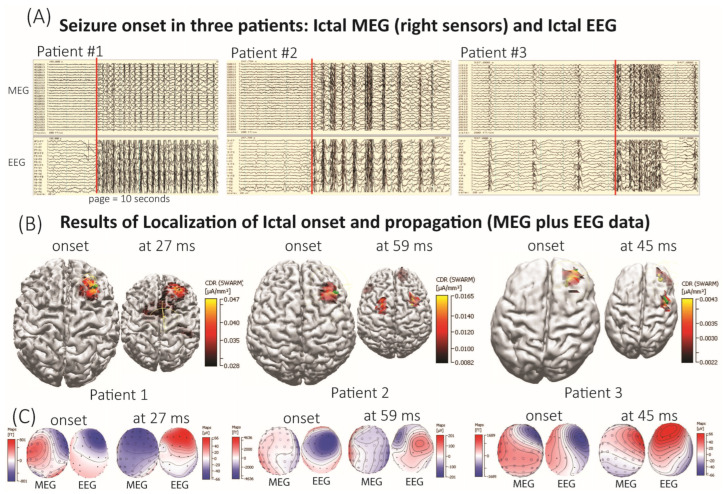
**CURRY Analysis of Generalized Seizure Onset Using Combined MEG and EEG Data**. (**A**) shows simultaneously recorded MEG and EEG traces of a generalized seizure with preceding normal background activity in patients #1 and #2. The red bar indicates the seizure onset, independently determined by an epileptologist (SP) and a neurophysiologist (VG). To see the time point of the localized ictal onset, please refer to Figure 2A (z-score graph) around the 0 s time mark. The MEG channels were from the right side of the MEG helmet. (**B**) displays the results of seizure onset localization and its rapid propagation in all three patients. The CURRY localization results overlay two different methods: equivalent current dipole (ECD; green arrows) and SWARM (red-yellow color gradient). The localization results are displayed on each patient’s individual 3D brain reconstruction derived from T1-weighted MRI. The cortical surface was rendered transparent at 60% to allow visualization of the underlying structure, and the threshold for the SWARM was at 50%. The results of seizure propagation are shown at the time points that are most relevant for both the ECD and SWARM analyses. Notably, the onset of the generalized seizure localizes to cortical areas in all three patients. (**C**) Illustrates EEG and MEG contour field maps (head top view) corresponding to the onset of generalized ictal activity and its propagation.

**Figure 2 brainsci-15-00938-f002:**
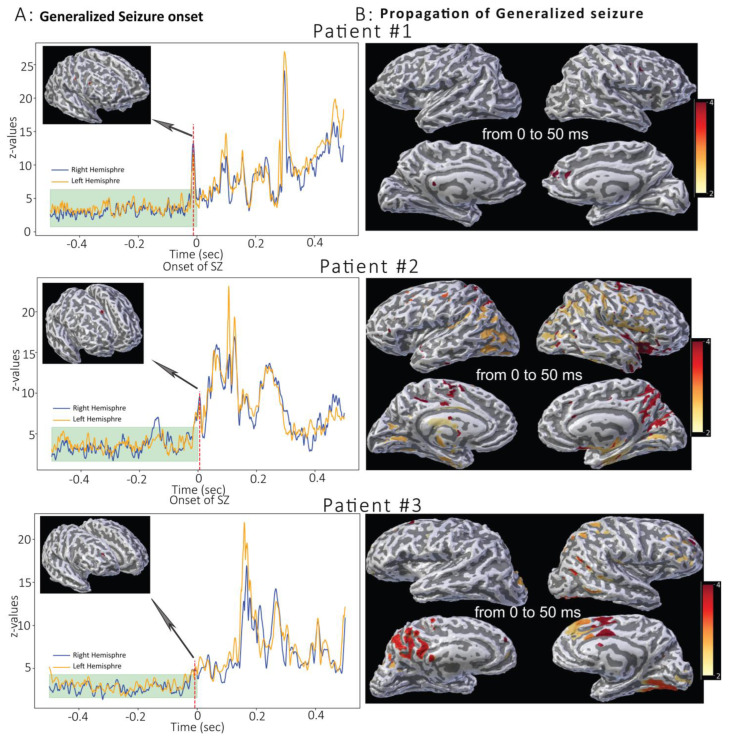
**MMVT Analysis of Generalized Seizure Activity in Three Patients.** (**A**) shows the graph of z-values representing normalized brain activity differences (sleep–seizure) at each millisecond within a 1 s time window. Time at 0 s indicates the seizure onset determined through both visual inspection and computational comparisons of brain activity across the left (orange line) and right (blue line) hemispheres. The cyan band (−0.5 s to 0 s) shows the baseline mean ± 2 SD. Under Gaussian assumptions, this ≈95% interval times where |z| > 2 fall outside this band. The dashed vertical red line indicates the time at which the maximum z-value in one hemisphere was significantly greater than in the other. The 3D image shows the onset of the generalized seizure in all three patients. (**B**) Illustrates the propagation, from 0 to 50 ms, of the generalized seizure on a 3D brain model for each patient. Z-values were mapped over a 50 ms window from the onset (Time = 0 s) in all three patients. Propagation is color-coded: red indicates seizure onset (0 s), transitioning to yellow at 50 ms latency. Propagation of the z-values over a 50 ms time window is measured from the refined onset time, where the activation time is plotted using a red-to-yellow gradient. Red = onset latency and yellow = 27 ms, 59 ms, 45 ms latency, respectively. As seen in (**A**), the onset of the generalized seizure is associated with the cortical frontal regions. In contrast, the propagation map in (**B**) shows a distributed network that includes subcortical areas, especially in patient #2 and patient #3. Notably, seizure onset does not involve subcortical regions.

**Figure 3 brainsci-15-00938-f003:**
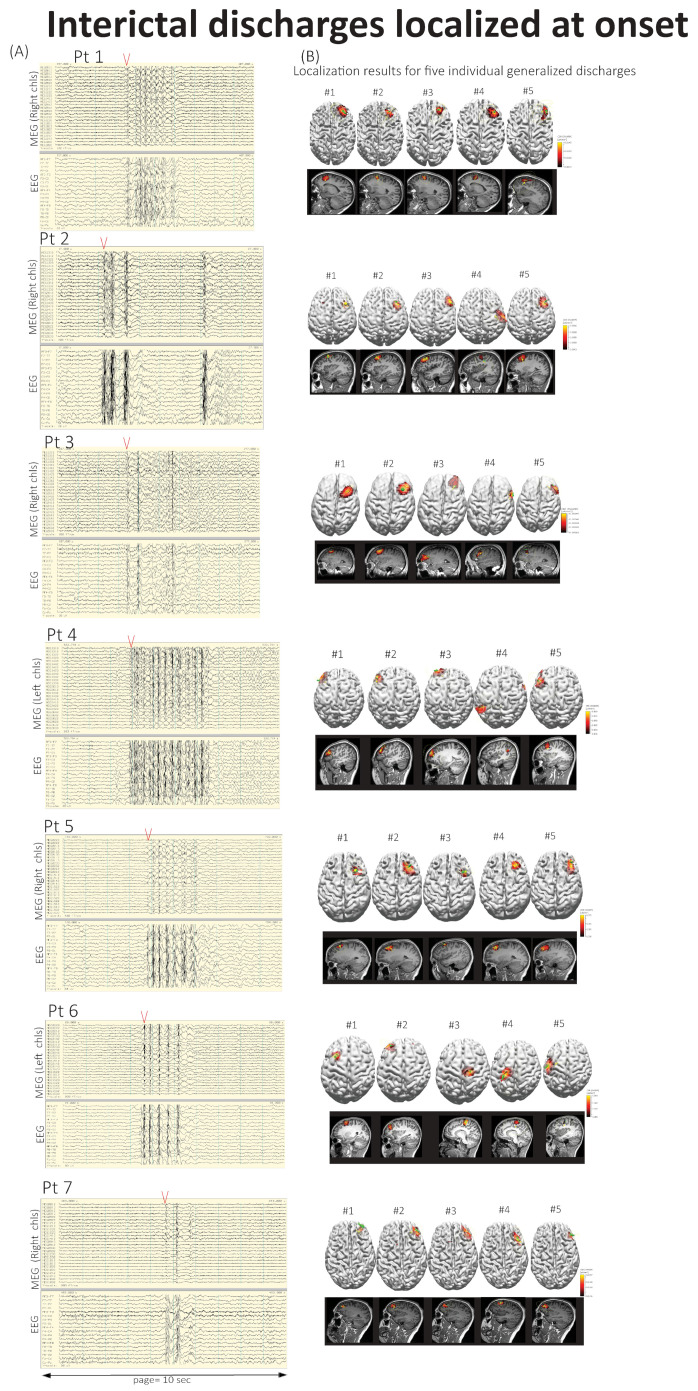
**Analysis of generalized interictal discharges in seven patients.** (**A**) An example trace showing simultaneously recorded MEG and EEG data. (**B**) Localization results for five individual generalized discharges. Two independent methods within the CURRY software, ECD and SWARM, were used for the source localization analysis. The source reconstruction solution was constrained to a 3D grid to avoid limiting the model to the cortex.

**Figure 4 brainsci-15-00938-f004:**
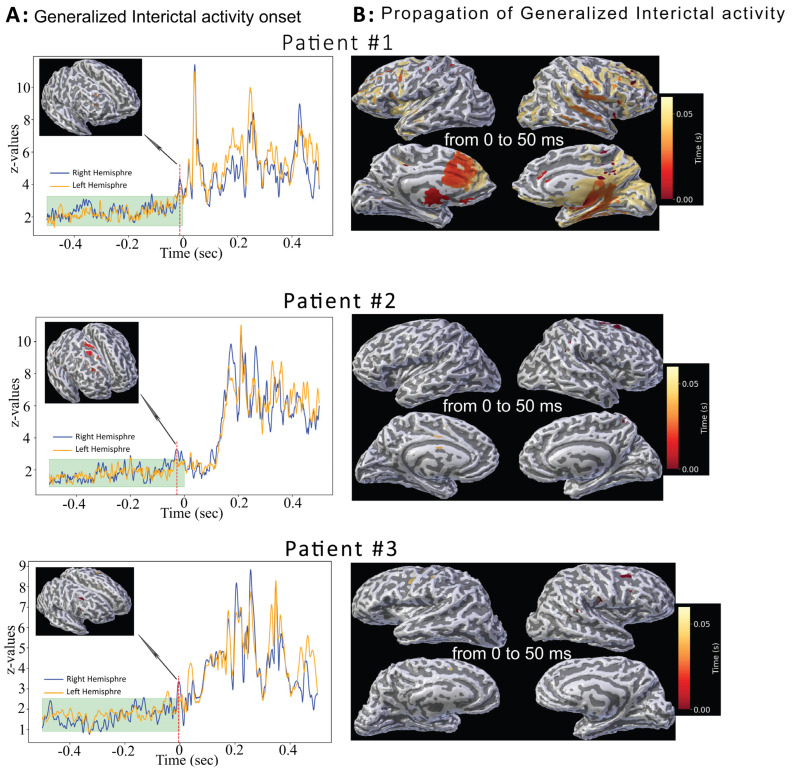
**MMVT Analysis of Generalized Interictal Activity in Three Patients.** Epochs associated with interictal activity were averaged for each patient. (**A**) shows the averaged graph of z-values representing normalized brain activity differences (sleep–interictal) at each millisecond within a 1 s time window. Time at 0 s marks the onset of the interictal discharge identified through both visual inspection and computational comparisons of brain activity across the left (orange line) and right (blue line) hemispheres. The cyan band (−0.5 s to 0 s) represents the baseline mean ± 2 SD (≈95% interval under Gaussian assumptions). The dashed vertical red line indicates the time (in milliseconds) when the maximum z-value in one hemisphere was significantly greater than in the other. Notably, the onset of generalized interictal discharges involves similar brain regions as those observed during ictal onset (see Figure 2) in all three patients. (**B**) Illustrates the propagation of generalized interictal discharges on a 3D brain model for each patient. Z-values were mapped, from onset to 50 ms, and identified both visually and computationally in all three patients. Propagation is color coded: red indicates the onset of the interictal discharge (0 sec), transitioning to yellow at 50 ms latency. Patient #1 shows rapid propagation of interictal activity involving subcortical regions.

**Table 1 brainsci-15-00938-t001:** Demographic, clinical characteristics of patients diagnosed with generalized epilepsies and clinical results.

**Pt**	**Age (Years)/Sex**	**Seizure Types**	**Medications**	**Imaging Results**	**Surface EEG Results**	**Clinical MEG Results**
1	16/F	Type 1: Absence Type 2: GTC	Diamox, Clobazam, Diazepam, Lorazepam	MRI: Normal PET: Decreased signal in the right frontal and left temporal	Occasional generalized spike-wave discharges with right frontal onset	Right frontal for ictal and interictal discharges
2	13/M	GTC	Depakote, Lamotrigine and Clobazam	MRI: Stable pineal region cyst PET: Normal	Generalized spike-and-wave discharges with slight right frontal predominance	Right frontal for ictal and interictal discharges
3	33/M	Bilateral clonic	Divalproex, Felbamate, Perampanel, Rufinamide	MRI: Normal PET: Left parietotemporal hypermetabolism	Generalized seizure pattern with right predominance	Right frontal for ictal Right fronto-centro-parietal for interictal discharges
4	13/F	Type 1: Absence Type 2: GTC	Keppra, Lamotringine	MRI: Lesion left temporal white matter PET: Hypometabolism R fronto-parietal	Absence seizure recorded maximum frontocentral	Defused bilateral frontal
5	12/M	GTC	Keppra	MRI: Normal PET: Normal	Generalized interictal activity with no lateralizing features	Right frontal
6	14/M	GTC	Depakote, Lamotringine	MRI: Normal PET: N/A	Generalized seizures and interictal activity with no lateralizing features	No cluster found
7	28/M	Bilateral convulsive seizures	Vimpat, Lamictal XR, Xcopri, Nayzilam	MRI: Normal PET: Decrease signal in left mesial temporal lobe	Focal and bilateral tonic clonic possible left frontal	Right frontal

**Table 2 brainsci-15-00938-t002:** Concordance between localization of generalized seizure onset and generalized interictal discharges onset.

**Pt #**	**CDR: Dipole Moving and SWARM**	**Brain Structure (Talairach Atlas)**
1	Ictal—Right frontal Interictals—Right frontal	R-Superior Frontal gyrus
2	Ictal—Right frontal Interictals—Right frontal	R-Middle Frontal gyrus
3	Ictal—Right frontal Interictals—Right frontal	R-Middle Frontal gyrus

## Data Availability

The data that support the findings of this study are available from the corresponding author, Valentina Gumenyuk, upon reasonable request. Public sharing of the data is restricted due to privacy concerns related to the patients who participated in the study.

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
