# Peer review of "Ictal MEG-EEG Study to Localize the Onset of Generalized Seizures: To See Beyond What Meets the Eye"

_brainsci, 2025, doi:10.3390/brainsci15090938_

Round 1

Reviewer 1 Report

Comments and Suggestions for Authors

This is an interesting manuscript on the ability of HDEEG and MEG to identify possible cortical substrates of early ictal activity in 3 patients who happened to experience 'generalised' seizures during their simultaneously acquired EEG (64 channel cap) and MEG (306 sensor array) recording.

All three patients had early localisation in the region of the right frontal lobe before apparent propagation and generalisation of the EEG/MEG signals.

I think the manuscript highlights the value of performing source imaging on the earliest phase information of EEG and MEG discharges as propagation can occur very rapidly. This is commonly underappreciated with standard low density EEG naked eye interpretation as much higher temporo-spatial sampling (HDEEG/MEG) is far more likely to be able to detect this transition phenomenon.     

I have a few issues with the manuscript however in its current form.

  1. I think the paper is talking to primary generalised epilepsy rather than secondary generalised epilepsy. But it might be talking about both. I think this needs to be made more obvious early. To be sure, some patients with IGE are later reclassified as focal epilepsy with secondary generalised discharges, particularly when higher density recordings assist with this distinction. Then again, the Authors also seem to be arguing that their results support a focal onset for patients who might otherwise be labelled as IGE. So this distinction needs to be fleshed out more, rather than simply referring to these cases as 'generalised epilepsy' and 'generalised seizures' as a catch all phrase. The main clinical value of MEG and HDEEG is for more accurate surgical localisation and I do not think the Authors are insinuating that refractory IGE can be addressed surgically?
  2.  None of the patients underwent ICEEG, so it is unclear what the ground truth might be in these cases, even though it is well known that ICEEG is not a true gold standard given the sampling problem.
  3.  The use of MEG and HDEEG as complementary modalities to inform the source generator(s) is valuable but there is no agreement on how these 'mixed' signals should be modelled. The main issue here is the difference in volume conductor properties for HDEEG vs MEG as tissue planes affect EEG signal detection more than MEG. This must somehow be factored into the equations for forward modelling. I do not see that the Authors have tried to do this or if they have appreciated this problem.
  4. The value of HDEEG is limited by the amount of sampling 'below the ears' to cover the basal surfaces of the frontal and occipital lobes and a lot of the temporal lobe. How much 'inferior chain' sampling is done with the 'EasyCap'?
  5.  What is meant by " seizure and interictal discharge onset were marked as the "first significant deflection" from baseline in the MEG and EEG traces? What constitutes a "significant deflection"?
  6. How were SNR values calculated for each epoch?
  7. What tissue conductivity values were used for the BEM modelling?
  8.  As with all manuscripts centred on source imaging the proof of the pudding is in the clarity and self-explanatory nature of the images. Fig 1A does not clearly show the signal 'significant deflection' for EEG and MEG that the Authors have decided on as ictal onset. The corresponding EEG and MEG contour field maps also need to be shown. Fig 1B also needs to include the field maps for early and later SWARM solutions. The CDR units are not shown clearly. The legend states that the colour bar shows the statistical values. This is not true. Although it is based on a probabilistic solution (sLORETA), SWARM is an expression of dipole current per volume (usually microAmp/mm2) so this should be corrected. 
  9. What threshold setting was used for the CDR map display?
  10. Note typos "Error! Reference source not found" and  "cortical focus theory" in bold throughout the manuscript. 
  11.  Figure 2 legend seems to use 50ms and 0.5s interchangeably...Should it be 500ms (as 0.5s) and not 50ms unless I am misunderstood?
  12. I find the images too hard to see in the left panel Fig 2. Please enlarge a little
  13. While the Fig 2 right panel supports right frontal ictal onset in Pt 1, the ictal onset in Pt 2 and 3 appears to be more widespread and even bilateral based on the red colouration in the images. Please clarify this.
  14.  I have the same comments for Fig 3 as per point 8 above. It is not clear which part of the deflection is used as a single time-point, how noise was determined, and what the corresponding field maps look like. Lumping EEG and MEG together in the same signal space is a simplification as outlined in point 3 above.
  15. For Figure 4, Pt 1 interictal activity appears to be more widespread and bilateral at onset so this should be clarified.
  16. The Discussion talks about generalised epilepsy and the need for EEG/MEG. Again, are the Authors suggesting IGE patients should be routinely recorded?
  17. Lack of apparent localisation to the thalamus does not equate to proof of cortical onset. Rather, it might be explained by modelling error so this needs to be appreciated.
  18. Discussion suggests that the study shows two distinct networks for interictal and ictal activity but this is not clear to me in the Results. Please elaborate. 
  19. How do the Authors know that the nodes influencing the interictal state are localised to the default mode network in these cases?
  20. I think the final paragraph is too broad and too sweeping and it is not adequately supported by the evidence given in the manuscript.         

Reviewer 2 Report

Comments and Suggestions for Authors

This study investigates the use of both EEG and MEG data to localize the onset of epileptiform discharges in patients with generalized epilepsies. While the simultaneous use of EEG and MEG is not novel per se, the application of this combined modality to generalized epilepsy is a significant and original contribution to the field. Previous studies have primarily focused on focal epilepsies or on characterizing various epileptiform activity such as interictal spikes and high-frequency oscillations, without necessarily addressing onset localization in generalized cases.

The central question addressed by this study is whether epileptiform discharges in patients with generalized epilepsies can be localized to a brain focus. This challenges the traditional view that such discharges involve the entire brain and therefore, do not lend themselves to source localization. Despite the limited sample size, the study presents compelling evidence that supports the concept of localizable onsets of both ictal and interictal discharges, thereby contributing to the growing support for the cortical focal theory.

This research is highly relevant to the field of epilepsy, from both a clinical and mechanistic standpoint. Improved understanding of the onset zones in generalized epilepsies has potential implications for diagnosis, treatment plans, and surgical candidacy in this patient population.

The authors provide a clear description of the methodology, including the use of sleep recordings as baseline for MEG analysis, which is appropriate and strengthens the interpretability of the results. The integration of EEG and MEG data is also a strength of the study.

The manuscript is presented in a logical and readable manner. The figures are generally clean and support the main points. The cited references are appropriate and include relevant prior work in the field.

The authors’ conclusions are consistent with the data and arguments presented. They successfully demonstrate that even in generalized epilepsies, a localized onset may be detectable using EEG and MEG. These findings may contribute to rethinking how we conceptualize generalized epilepsies and their potential treatment avenues.

However, the study would benefit from addressing the following points:

  1. In Figure 3(A), it would be helpful to include the contralateral hemisphere’s EEG and MEG traces during the same 10-second epoch. For example, in Pt 1, what is observed in the left hemisphere during the interictal discharge shown? This would provide insight into hemispheric symmetry or asymmetry, which is central to the localization claim.
  2. There is an inconsistency in the labeling of confidence intervals. "Cyan" is used in the Methods section, whereas "green" is used in the Results section and figure legends. Please ensure consistent terminology throughout the manuscript.
  3. On page 6, line 198 and again on page 8, line 229, the text reads: "Error! Reference source not found." This appears to be a formatting artifact and should be corrected.
  4. At the top of Figure 3 and directly below the label "(B)", the word "results" is misspelled.
  5. In the same figure, the red tick mark indicating onset is missing for Pt 7.

Overall, this is a thoughtful and well-designed study. With minor revisions and clarifications as suggested above the manuscript will make a valuable contribution to the field of epilepsy.

Reviewer 3 Report

Comments and Suggestions for Authors

In this manuscript, the authors investigated the utility of simultaneous magnetoencephalography (MEG) and electroencephalography (EEG) in localizing the onset of generalized seizures in patients with refractory generalized epilepsy. The study included seven patients, three of whom had typical generalized seizures during the MEG recording. Using multiple source localization methods - equivalent current dipole (ECD), sLORETA, and dSPM - the authors demonstrated that seizure and interictal discharge onset could be localized to cortical areas, with consistent findings across methods. The results support the cortical focus theory and suggest that MEG-EEG source imaging may have broader applications in generalized epilepsy. This is an important and innovative study that challenges prevailing assumptions and expands the scope of noninvasive neuroimaging in epilepsy. After thoroughly reviewing the manuscript, I believe the paper would benefit from a major revision. Please see my specific comments below.

  • The sample size is small, with only three patients contributing ictal events. Please acknowledge this limitation more explicitly and discuss how it may affect the generalizability of the findings.
  • The selection criteria for the seven patients are not fully described. Were these consecutive referrals? Were any patients excluded (e.g., due to motion, artifact, or lack of seizures during recording)? Please clarify to avoid selection bias.
  • Statistical rigor can be improved. While z-score thresholds are reported, no confidence intervals are provided for localization metrics, and intersubject variability in onset localization is not fully discussed. Please consider including additional quantitative measures of localization accuracy or reproducibility.
  • The use of MMVT and CURRY in tandem is commendable. However, it would be helpful to elaborate on how discrepancies between these tools, if any, were resolved or interpreted.
  • Please elaborate on how “focal” onset was defined—especially in the context of a generalized seizure phenotype. For instance, how was the threshold for lateralization or regional dominance determined?
  • Clinical implications should be discussed in more detail. Do the authors recommend broader use of MEG in generalized epilepsy patients, and if so, in what clinical scenarios (e.g., drug resistance, unclear diagnosis)? How might these findings influence patient management?
  • The manuscript would benefit from careful proofreading. There are several grammatical inconsistencies and awkward phrasings throughout the text. Additionally, please ensure that all abbreviations are defined upon first use.
  • Please clarify whether seizure onset was ever localized to subcortical structures (e.g., thalamus). The abstract states that all localizations were cortical, yet the discussion includes subcortical propagation. This distinction is important.
